# Versatile seamless DNA vector production in *E. coli* using enhanced phage lambda integrase

**Suki Roy, Sabrina Peter, Peter Dröge** *

School of Biological Sciences, Nanyang Technological University, Singapore, Singapore

* pdroge@ntu.edu.sg

## Abstract

Seamless DNA vectors derived from bacterial plasmids are devoid of bacterial genetic elements and represent attractive alternatives for biomedical applications including DNA vaccines. Larger scale production of seamless vectors employs engineered *Escherichia coli* strains in order to enable tightly regulated expression of site-specific DNA recombinases which precisely delete unwanted sequences from bacterial plasmids. As a novel component of a developing lambda integrase genome editing platform, we describe here strain *MG1655-ISC* as a means to easily produce different scales of seamless vectors, ranging in size from a few hundred base pairs to more than ten kilo base pairs. Since we employed an engineered lambda integrase that is able to efficiently recombine pairs of DNA crossover sites that differ in sequence, the resulting seamless vectors will be useful for subsequent genome editing in higher eukaryotes to accommodate variations in target site sequences. Future inclusion of single cognate sites for other genome targeting systems could enable modularity. These features, together with the demonstrated simplicity of *in vivo* seamless vector production, add to their utility in the biomedical space.

## Introduction

Seamless DNA vectors—also referred to as "minicircles or minimized vectors" [1, 2]—are derived from bacterial plasmids and devoid of bacterial genetic elements such as origins of replication and resistance markers. Seamless vectors are circular, covalently closed and usually negatively supercoiled DNA molecules. They are becoming increasingly attractive for biomedical applications such as cell line engineering, biologics production, gene/cell therapy and DNA vaccination. This attractiveness is due to their enhanced transgene expression, improved safety features, reduced gene silencing, and higher gene transfer efficiency when compared to parental plasmids [3, 4].

The most common means to produce seamless DNA vectors are site-specific DNA recombinases which utilize their cognate DNA sequences that flank unwanted bacterial genetic elements. Through a precise DNA strand cutting and pasting reaction at their respective cognate sites, these enzymes can splice out intervening DNA from the rest of the molecule. The

**Data Availability Statement:** All relevant data are within the paper and its Supporting Information files.

**Funding:** This work was supported through a grant from the National Research Foundation-Competitive Research Programme, Singapore to P.

D. (NRF-CRP21-2018-0002). The funders had no role in study design, data collection and analysis, decision to publish, or preparation of the manuscript.

**Competing interests:** P.D. is a founder and stakeholder of LambdaGen Therapeutics Pte Ltd. S. R., S.P. and P.D. are listed as co-inventors on a related patent application.

recombination reaction using plasmids as substrate thus results in two circular DNA molecules: one that carries the unwanted bacterial elements and the other representing the desired seamless vector. The site-specific recombination reaction can be carried out either *in vitro* with purified enzymes for cost-effective small scale production or *in vivo* inside the bacterium *Escherichia coli* to achieve medium to large scale production; the latter is currently being employed in a few commercial settings. Various protocols are available to isolate and purify seamless vectors for downstream purposes at industrial scales [5].

A number of recombinases have been utilized for *in vivo* production of seamless vectors in *E. coli* and include the wild-type phage lambda integrase [6], the yeast recombinases Cre and FLP [7, 8], the wild-type phage ΦC31 integrase [9], and the ParA resolvase [10]. For *in vivo* seamless vector production, a technical challenge is the stringent control of the expression of the recombinase. Leaky expression during bacterial growth will result in premature loss of the seamless vector inside the bacterium and hence severely compromises yields. In order to achieve stringent recombinase expression control, several expression systems have been employed including temperature-sensitive lambda repressor cI857/pR [6] and the plasmid-based pBAD/araC arabinose system [9]. However, employing these systems reproducibly and at larger scales remains technically challenging due to low recombination efficiency resulting in less yield of seamless vector and contamination with parental plasmids in the final preparation of seamless vectors [11, 12].

In the present study, we present an engineered *E. coli* strain that can easily be used for multi-scale and multi-purpose seamless vector production by *in vivo* site-specific recombination catalyzed by an enhanced mutant lambda integrase, dubbed IntC3 [13]. Distinct advantages of this system are the ease of handling and, due to the employment of enhanced IntC3, the application of various pairs of attachment (*att*) recombination site derivatives as efficient recombination substrates to yield seamless vectors ranging in size between a few hundred base-pairs to > 10 kb. This flexibility in *att* site sequences considerably expands the scope of future downstream applications for seamless vectors including site-specific genome editing of higher eukaryotic cells [14, 15].

## Materials and methods

### Engineering of *E. coli* strain *MG1655* carrying an inducible IntC3 expression cassette

The *E. coli* strain *MG1655* was chosen as a base for generating a versatile seamless vector bacterial producer strain because it approximates K12 wild-type cells with minimal prior genetic changes [16]. We hypothesized that this feature provides us with higher chances of success for tightly regulated expression of IntC3 via the endogenous *arabinose* (*ara*) operon. *MG1655* has been maintained as a laboratory strain with minimal genetic manipulation, having only been cured of the temperate bacteriophage lambda and F plasmid by means of ultraviolet light and acridine orange, respectively [17, 18].

A policistronic sequence of 2725 bp, termed ISC, composed of the *IntC3*, single chain integration host factor *scIHF2* [19] and chloramphenicol resistance gene (*CAT*) was commercially synthesized (GeneScript) (**S1 Fig**). scIHF2 (single polypeptide chain IHF) is a modified version of wild-type IHF composed of the alpha subunit of IHF inserted into the beta subunit through a novel protein engineering approach. It exhibits similar properties to wild-type IHF in term of DNA binding and bending [19]. The chosen strategy for *MG1655* engineering included the precise insertion of the ISC expression cassette into the *ara* operon of *MG1655* immediately downstream of the promoter by using the start codon of the endogenous *araB* gene as start codon for *INTC3* (**Fig 1A**). Two primers were designed to insert the construct at this locus

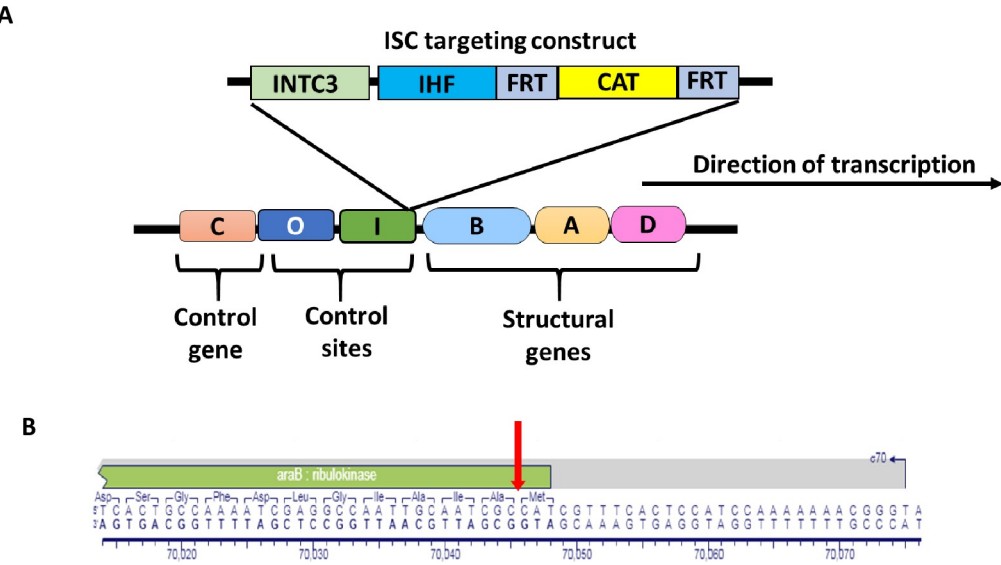

**Fig 1. Genetic modification of strain *MG1655-ISC*.** (**A**) Diagram of the *ara* operon with the inserted gene cassette composed of integrase variant C3 (INTC3), single chain integration host factor 2 (IHF), recombination sites for Flp recombinase (FRT) as direct repeats and chloramphenicol resistance cassette (CAT). The direction of transcription of the modified *ara* operon is indicated. (**B**) The red arrow demarcates the precise position of ISC insertion by homologous recombination between start and alanine codons of *araB* as verified by sequencing.

using routine protocols of *lambda* red-mediated homologous recombination reactions [20]. The ATG start codon of the *IntC3* gene was re-introduced in the forward primer (**Fig 1B**). PCR amplification of the ISC construct for electroporation into electroporation-competent *MG1655* cells was performed with primers: INTC3_ARAB_FWD_HR:ACTCTCTACTGTTT CTCCATACCCGTTTTTTTGGATGGAGTGAAACGATGGGAAGAAGGCGAAGTCATGAGC and FRT_ARAB_REV_HR: GCCAAAGCTCGCACAGAATCACTGCCAAAATCGAGGCCAATTGCAA TCGCTTATACAGTCGAAGTTCCTATA using Q5 High Fidelity DNA Polymerase (New England Biolabs). The thermal cycling parameters used for PCRs were as follows: initial denaturation at 98˚C for 30 seconds, 35 cycles of denaturation at 98˚C for 10 seconds, annealing at 65˚C for 30 seconds and extension at 72˚C for 2 minutes, and a final step of 72˚C for 2 minutes. The resulting purified PCR product was used for subsequent electroporation into *MG1655*/pKD46 electroporation competent cells described below.

*MG1655*/pKD46 cells were made competent for plasmid transformation using standard protocols. Briefly, plasmid pKD46 [21] was transformed, cells plated on ampicillin selection media, and grown at 30˚C overnight. A single colony of pKD46-transformed *MG1655* was grown overnight in DYT media plus ampicillin (200μg/ml) at 30˚C. The stationary culture was diluted 1:200 in fresh 200 ml DYT media and incubated at 30˚C. When $OD_{600}$ had reached 0.4, the culture was induced with 1.2% L-arabinose and incubated at 37˚C for 1 hr at 180 rpm. Cells were immediately chilled in an ice water bath. The culture was left on ice for 20 minutes with occasional agitation. Subsequently, cells were kept chilled and distributed into 4 X 50 ml Falcon tubes on ice. Centrifugation was at 1000g for 10 to 20 minutes at 4˚C. Once the supernatant became clear, cell pellets were suspended in 25 ml of double-deionized (DD) ice-cold water. Contents of 2 X 50 ml Falcon tubes were combined and centrifuged again under the same conditions. After the supernatant became clear, each pellet was suspended in 50ml ice cold water and centrifugation was repeated. The supernatant was discarded and the pellet suspended in 50 ml ice cold 10% glycerol (diluted in DD water). The centrifugation step was

repeated and the pellet re-suspended in 0.5 ml 10% glycerol. The two re-suspended pellets were combined in 2 ml pre-chilled Eppendorf tubes and mixed. $OD_{600}$ of a 1:100 diluted solution (10 μl suspended cells + 990 μl of 10% glycerol) was set 0.4 to 0.6. Competent cells were stored in aliquots of 70μl-100μl per tube after flash freezing in liquid nitrogen and stored at minus 80˚C.

Electroporation of the ISC PCR-amplified construct into electroporation competent *MG1655*/pKD46 cells was performed with Gene Pulser (BioRad) as follows: 1 to 10 ng of the PCR product were added to 100 μl competent *E. coli* and electroporated in pre-set conditions (set 1 or 2). Cell recovery was at 37˚C for 1 hr in DYT without antibiotics. The transformed cells were spread onto DYT media + 0.1% Glucose + 15 μg/ml chloramphenicol agar plates and grown at 30˚C. Growth at 37˚C will subsequently lead to the loss of *pKD46* plasmid since it carries a temperature sensitive origin of replication. Resulting colonies were tested by colony PCR to confirm the left and right junctions of the inserted transgene cassette, as well as the presence of the entire cassette by genomic PCR (**S2 Fig**).

Primers for left junction PCR resulting in a 291 bp product: ARAC_FWD GTCTATAATCA CGGCAGAAAAGTCC and INTC3_REV TCGCCTGTCTCTGCCTAATCC. Right Junction primers will yield a PCR product of 397 bp: CAT_FWD CGCAAGGCGACAAGGTGCT and ARAB_REV CCGCTTCCATTGACTCAATGTAGTC. Genomic PCR using ARAC_FWD and ARAB REV primers will generate a 2.85 kb product.

Colony PCR was performed as follows: One colony was diluted in 50 μl DYT media + 0.1% Glucose + 15 μg/ml chloramphenicol media and 2 μl used as colony PCR template. PCR was performed using GoTaq Flexi DNA polymerase (Promega) to amplify both the junctions in 25 μl reactions. The thermal cycling parameters used for PCRs were as follows: initial denaturation at 95˚C for 5 minutes, 35 cycles of denaturation at 95˚C for 1 minute, annealing at 56˚C for 30 seconds and extension at 72˚C for 1 minute, and a final step of 72˚C for 5 minutes. PCR products were verified by sequencing. We dubbed our engineered strain *MG1655-ISC*.

## Results

### Growth of engineered *MG1655* strains carrying integrase expression cassettes

Our *MG1655-ISC* strain retained at this stage the CAT gene as part of the transgene cassette. By including the FRT sequences for the yeast Flp recombinase (**Fig 1**), the option remained to remove the CAT expression cassette by transient Flp recombinase expression at a later stage. A critical parameter of any engineered *E.coli* strain for seamless vector production is the cell doubling time. We analyzed growth rates of *MG1655-ISC* and compared it with that of another engineered *MG1655* strain that carries the same transgene expression cassette at the *ara* locus, except that *scIHF2* [19] had been omitted. The latter strain was dubbed *MG1655-IC* and had been generated in parallel to *MG1655-ISC* following the same protocol.

An example of the growth rate analysis (**S3 Fig**) revealed that the presence of both*IntC3* and *scIHF2* in the *ara* operon had no substantial effects on exponential growth rates, with cell doubling times in the typical range of 20 to 30 minutes. Furthermore, the presence of *scIHF2* does not change the cell density at stationary phase.

### Outline of multi-scale seamless vector production using *MG1655-ISC*

The general workflow for seamless vector production using strain *MG1655-ISC* is summarized in **Fig 2**. A plasmid with a standard bacterial backbone is flanked by two directly repeated lambda integrase recombination sequences, termed *att1* and *att2*, and carries the desired DNA

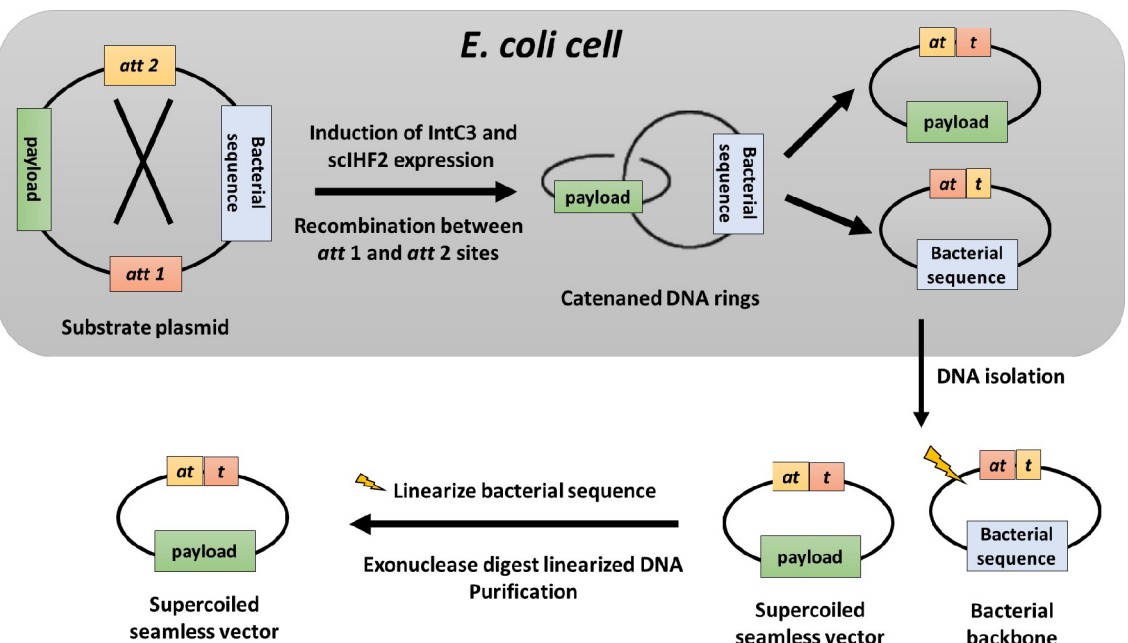

**Fig 2. Outline of multi-scale seamless vector production using *MG1655-ISC*.** See text for details. The catenated DNA rings generated by recombination inside *E. coli* are efficiently resolved into two monomeric DNA circles by topoisomerase IV. The DNA circles were purified from the *E. coli* strains and digested with an appropriate restriction enzyme to linearize the bacterial backbone and unrecombined substrate plasmid. This was followed by exonuclease treatment to digest the linear and nicked circular DNA products. Hence, only supercoiled seamless vector will be retained.

payload for seamless vector production. After transformation into *MG1655-ISC*, cells were grown with antibiotics until $OD_{(600)}$ reached 1.0. IntC3 and scIHF2 expression was induced by the addition of arabinose (1.5% final concentration), and cells were incubated for additional 70–90 minutes at 37˚C.

Induction of IntC3 leads to recombination between *att1* and *att2* which generates a dimeric DNA catenane consisting of one DNA ring that carried the bacterial backbone and a second DNA ring that carried the DNA payload plus one copy of a hybrid *att* sequence. The catenated DNA rings can be efficiently unlinked inside *E.coli* by endogenous type 2 topoisomerases.

Following induction of transgene expression, episomal DNA is purified from lysed *MG1655-ISC* cells by standard procedures, and the circular bacterial DNA is linearized by restriction digestion. The linearized DNA and contaminating nicked DNA molecules can be effectively degraded with phage *T5* exonuclease. The remaining DNA is the intact, covalently closed, supercoiled seamless vector.

## Mini-seamless vector production using variants of *att*L and *att*B sites in *MG1655-ISC*

To demonstrate broad utility of *MG1655-ISC* for seamless vector production, we first transformed the 6.3 kb substrate plasmid p*att*Phae2 (*att*L) (**Fig 3A**). This recombination substrate carried a 21 bp *att*B variant and a 121 bp *att*L variant recombination sequence in direct repeat orientation separated by about 530 bp. Recombination by IntC3 will result in two circular DNA molecules: a small supercoiled 530 bp mini-seamless vector and 5.8 kb supercoiled DNA that carries the bacterial backbone and other sequences.

*MG1655-ISC* cells transformed with p*att*Phae2 (*att*L) were induced by arabinose for 70 minutes and harvested. Episomal DNA was isolated and analyzed by agarose gel

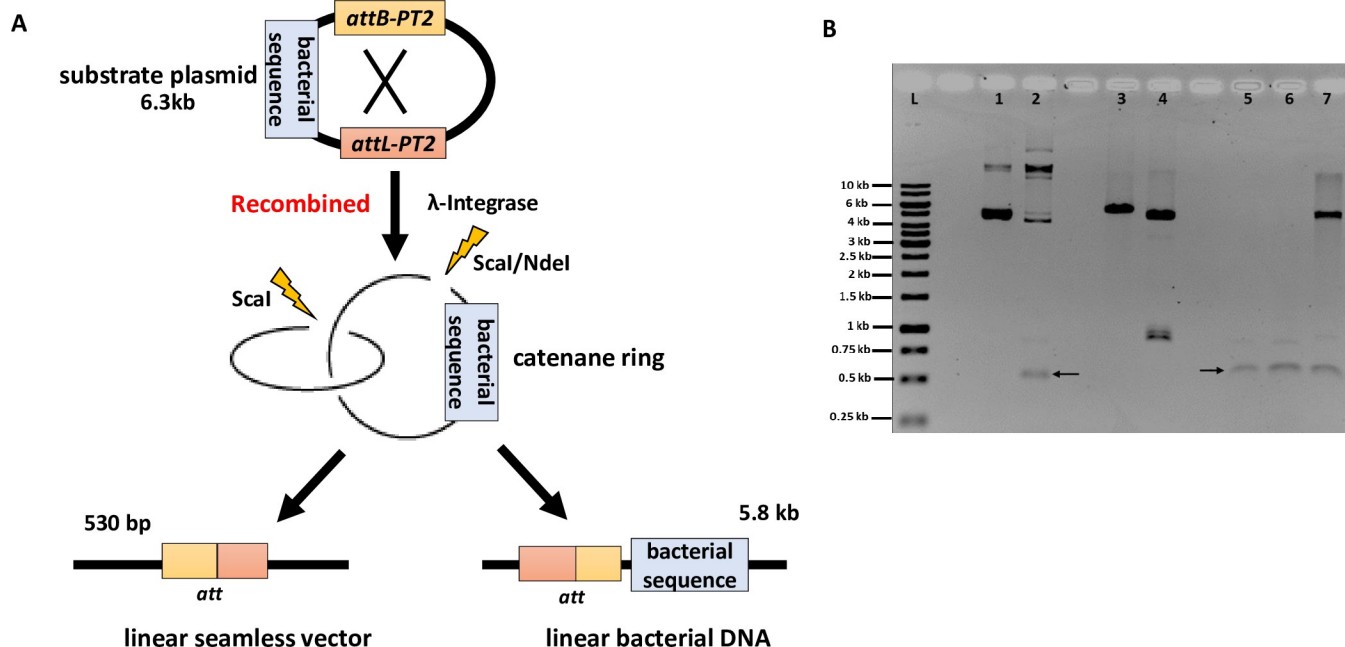

**Fig 3. Production of mini-seamless vectors in *MG1655-ISC*.** (**A**) Map of recombination substrate p*att*Phae2 (*att*L). The *att*B and *att*L (*PT2*) sequences are derivatives from the corresponding wild-type *att*B and *att*L and were placed about 500 bp apart as direct repeats. Recombination deletes the DNA segment flanked by the two *att* sites. (**B**) A suspension culture (10ml) was induced for integrase expression by arabinose and further incubation for 70 minutes. Plasmid DNA was purified and analysed by agarose gel electrophoresis. Lane L: marker ladder; Lane 1: substrate vector *att*Phae2 (*att*L) undigested; Lane 2: substrate vector *att*Phae2(*att*L) after induction, undigested; Lane 3: substrate vector *Nde*I digested; Lane 4: substrate vector *Sca*I digested; Lane 5: purified plasmid DNA after arabinose induction was digested with *Nde*I and *T5* exonuclease; 3μl of 100μl total sample loaded; Lane 6: same as lane 5, 6 μl loaded; Lane 7: Purified plasmid DNA after arabinose induction digested with *Sca*I. Note that the supercoiled and linearized seamless vectors run at about the same position.

electrophoresis. The results (**Fig 3B**) revealed that compared to substrate DNA (lane 1), induction of IntC3 expression produced recombination products which were decatenated *in vivo* by endogenous topoisomerases, i.e., the two recombination product DNA rings were no longer physically (topologically) linked (lane 2; diagrammed in **Fig 2**). The released supercoiled mini-seamless vector (lane 2; arrow) migrates far ahead from the rest of the isolated DNA due to its small size. Restriction digestion of substrate DNA and recombination products by *Sca*I confirmed that the vast majority of the substrate had been recombined inside *MG1655-ISC* cells after arabinose induction (lanes 4 and 7, respectively). Restriction digestion of DNA isolated from induced cells by *Nde*I, which cleaves only the bacterial backbone segment, in the presence of *T5* exonuclease resulted in supercoiled mini-seamless vector DNA with only spurious amounts of non-supercoiled (presumably nicked) mini-seamless DNA (lanes 5 and 6; arrow). The substrate DNA digested with *Nde*I without nuclease treatment is analyzed in lane 3. We determined that about 3 μg of pure supercoiled mini-seamless vector can easily be produced from a 100 ml culture.

## Maxi-seamless vector production using variants of *att*P and *att*B sites in *MG1655-ISC*

In another example demonstrating broad utility of strain *MG1655-ISC* for seamless vector production, we transformed the 13.4 kb substrate plasmid *att*P4x*att*H4x (**Fig 4A**). This recombination substrate carried a 21 bp *att*B variant (*att*H4x) and a 241 bp *att*P variant (*att*P4x) [13]. Recombination by IntC3 will result in a large supercoiled 10.3 kb seamless vector that carries a

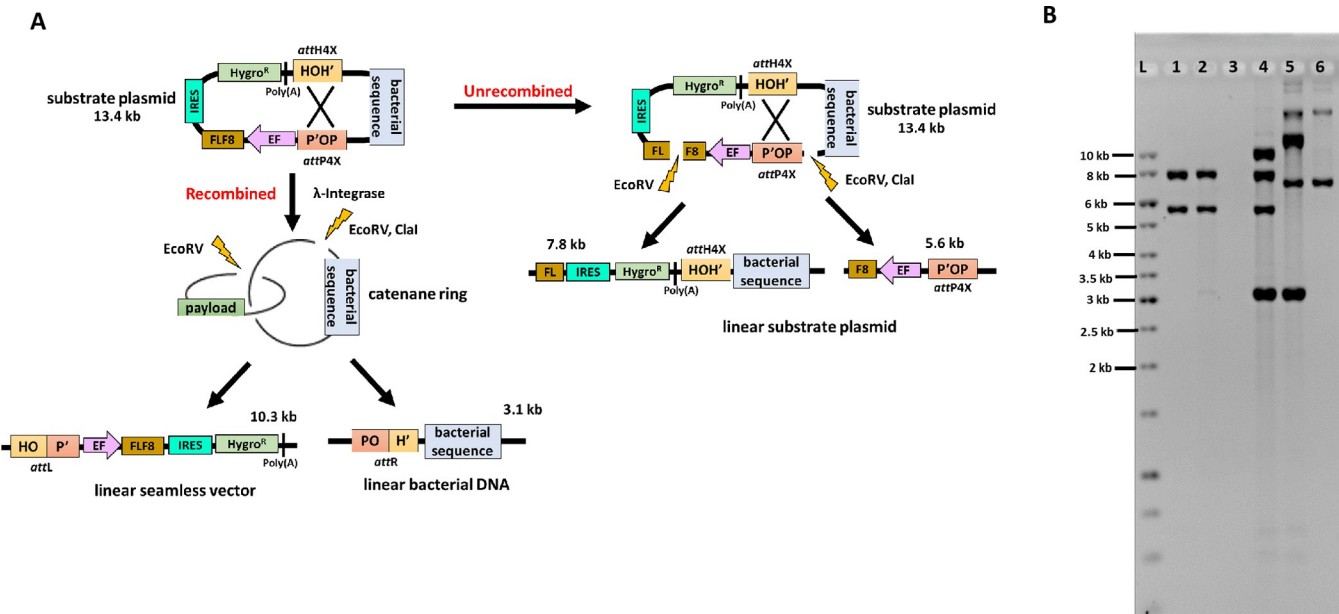

**Fig 4. Production of maxi-seamless vectors in *MG1655-ISC*.** (**A**) Map of recombination substrate p*att*P4x*att*H4x. The 21 bp *att*H4x and the 241 bp *att*P4x sequences are derivatives of the corresponding wild-type *att*B and *att*P sites. They flank the 3 kb bacterial backbone as direct repeats. Recombination deletes the bacterial backbone. Relative positions of relevant restriction sites are marked and predicted recombined and un-recombined restriction fragments indicated. (**B**) A suspension culture (10 ml) was induced for integrase expression by arabinose and further incubation for 70 minutes. Plasmid DNA was purified and analysed by agarose gel electrophoresis. L: 1 kb ladder; Lane 1: substrate vector (before transformation) digested with *Eco*RV generating two bands of size 7.8 kb and 5.6 kb for linear plasmid; Lane 2: transformed substrate vector (just before induction) digested with *Eco*RV generating two bands of size 7.8 kb and 5.6 kb for linear unrecombined plasmid; Lane 3: empty; Lane 4: substrate vector (after induction) digested with *Eco*RV generating four bands of size 7.8 kb and 5.6 kb for linear unrecombined plasmid, 10.3 kb for linear seamless vector and 3.1 kb for linear bacterial backbone; Lane 5: substrate vector (after induction) digested with *Cla*I generating three bands of size 13.4 kb for linear unrecombined plasmid, 3.1 kb for linear bacterial backbone and 10.3 kb for supercoiled seamless vector; Lane 6: substrate vector (after induction) digested with digested with *Cla*I and *T5* exonuclease generating a supercoiled seamless vector. Note that the additional high molecular weight DNA is most likely the result of intermolecular recombination between seamless vector generating dimers.

human blood clotting factor 8-Ires-hygromycin expression cassette plus a hybrid *att*L sequence. The second predicted recombination product is a 3.1 kb supercoiled circular DNA that carries bacterial genetic elements plus a hybrid *att*R sequence.

Transformed *MG1655-ISC* cells were induced by arabinose as before, and isolated plasmid DNA was analyzed by agarose gel electrophoresis for recombination products. Results obtained with *Eco*RV-digested DNA (**Fig 4B**) revealed that compared to the untransformed substrate DNA (lane 1), the vast majority of plasmid remained unrecombined after transformation to the time point just before induction of IntC3 expression by arabinose (lane 2), hence again demonstrating tight regulation of *IntC3* gene expression from the engineered *ara* operon under the chosen experimental conditions. Induction of IntC3 expression by arabinose produced a majority of the predicted 10.3 kb and 3.1 kb recombination products (*Eco*RV digest, lane 4). Restriction digestion with *Cla*I, which cleaves only in the bacterial DNA backbone, resulted in supercoiled 10.3 kb seamless vector, the linearized unrecombined substrate (13.4 kb) and the linearized bacterial backbone (3.1 kb) (lane 5). Moreover, the 10.3 kb seamless vector remained the only product after addition of *T5* exonuclease to the *Cla*I digest (lane 6). In this example, we also observed a possible small fraction of seamless vector dimers, which could be the result of intermolecular recombination between two hybrid *att*L sites occurring subsequent to intramolecular recombination (lanes 5 and 6). We determined that 60 to 90μg of supercoiled maxi-seamless vector can be produced from a 100ml culture under standard laboratory conditions after phenol/chloroform extraction and ethanol precipitation.

## Discussion

We presented here the engineering and application of a novel *E. coli* strain carrying a tightly regulated gene cassette for the expression of the lambda integrase variant IntC3 inserted into the *ara* operon. By using a simple cell culture protocol, we demonstrated that IntC3 very efficiently recombined plasmids inside *E. coli* to yield seamless vectors ranging in size between 500 bp and >10 kb. The employment of restriction enzymes together with the highly active phage *T5* exonuclease ultimately resulted in high yields of pure negatively supercoiled seamless vectors.

Novel lambda integrase variants have recently been generated as part of an evolving integrase platform technology that enables site-specific insertion of large transgene cassettes into predetermined endogenous or artificial human genomic *att* target sequences [13]. Since these genomic targets vary in sequence, it is important to be able to easily generate sufficient seamless target vectors that can accommodate these sequence variations in the corresponding vector-born *att* sites as recombination partners. The use of our catalytically enhanced IntC3 as recombinase inside *E.coli* addresses this issue. We demonstrated this by efficient recombination of distinct pairs of *att* sites on the substrate vectors, e.g. *att*BPhae2 x *att*LPhae2 and *att*H4x x *att*P4x. This sequence flexibility will expand the lambda integrase *att* sequence space of potential endogenous target site selection in higher eukaryotes beyond the human genome specifically for seamless payload vector integration. Furthermore, it will be easy to also include specific recombination or cleavage sequences for other genome editing systems on seamless vectors for modular use.

We have shown previously that functional seamless vectors can also be generated by purified integrase *in vitro* [15]. However, for applications that require large scale production, e.g., as DNA vaccines, producer *E. coli* strains need to be deployed [22]. Our novel strain *MG1655-ISC* presented here may thus become a useful means to achieve this objective based on the demonstrated ease in handling and the efficiency of recombination in the absence of leakiness. Furthermore, lambda integrase can recombine the wild-type *att*P site on an episomal vector with the genomic *att*B site in *E.coli* [23] which would lead to bacterial cell toxicity due to incompatible two origins on the genome. Hence, the ability of using variant *att* site sequences with strain *MG1655-ISC* may represent an advantage for large scale approaches.

## Supporting information

**S1 Raw image.**
(PDF)

**S1 Fig. Sequence of *ISC* target construct for generating *MG1655-ISC*.** See text for details.
(PPTX)

**S2 Fig. PCR confirmation of successful targeting events in *MG1655-ISC*.** (**A**) Colony PCR was performed with primers ARAC_FWD and INTC3_REV for left junction. PCR amplified products of the expected size 291 bp were detected in all colonies. (**B**) Colony PCR was performed with primers CAT_FWD and ARAB_REV for right junction. PCR amplified products of the expected size 397 bp were detected in all colonies. (**C**) Genomic PCR was performed with primers ARAC_FWD and ARAB_REV for full insertion amplification. PCR amplified products of the expected size 2.8 kb were detected in two colonies. L: 100 bp ladder; Lanes 1–4: *MG1655*/pKD46 colonies electroporated with ISC PCR-amplified construct.
(PPTX)

**S3 Fig. Growth curves of *MG1655-ISC* and *MG1655-IC*.** Cells of both strains in stationary phase were inoculated in fresh media and $OD_{600}$ measured at different time points as

indicated. Both strains show very similar exponential growth rates and cell densities in late stationary phase.
(PPTX)

## Acknowledgments

Special thanks to Dr. Michael Berger, University of Münster, Germany, for expert advice in *E. coli* strain construction. This work was supported through a grant from the National Research Foundation-Competitive Research Programme, Singapore to P.D. (NRF-CRP21-2018-0002). Funding for open access charge: National Research Foundation Competitive Research Programme, Singapore (NRF-CRP21-2018-0002). The funders had no role in study design, data collection and analysis, decision to publish, or preparation of the manuscript.

## Author Contributions

**Conceptualization:** Peter Dröge.

**Funding acquisition:** Peter Dröge.

**Investigation:** Suki Roy, Sabrina Peter.

**Project administration:** Sabrina Peter.

**Supervision:** Peter Dröge.

**Writing – original draft:** Peter Dröge.

**Writing – review & editing:** Suki Roy, Sabrina Peter, Peter Dröge.

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
