## [Decision Letter · Decision Letter 0]

25 Jul 2022

PONE-D-22-16259Versatile seamless DNA vector production in E. coli using enhanced phage lambda integrasePLOS ONE

Dear Dr. Dröge,

Thank you for submitting your manuscript to PLOS ONE. After careful consideration, we feel that it has merit but does not fully meet PLOS ONE’s publication criteria as it currently stands. Therefore, we invite you to submit a revised version of the manuscript that addresses the points raised during the review process.

We look forward to receiving your revised manuscript.

Kind regards,

Chen Ling, Ph.D.

Academic Editor

PLOS ONE

Journal Requirements:

"supported through a grant from the National Research Foundation-Competitive Research Programme, Singapore to P.D. (NRF-CRP21-2018-0002). Funding for open access charge: National Research Foundation Competitive Research Programme, Singapore (NRF-CRP21-2018-0002)"

3. We note that you have a patent relating to material pertinent to this article. Please provide an amended statement of Competing Interests to declare this patent (with details including name and number), along with any other relevant declarations relating to employment, consultancy, patents, products in development or modified products etc. Please confirm that this does not alter your adherence to all PLOS ONE policies on sharing data and materials, as detailed online in our guide for authors http://journals.plos.org/plosone/s/competing-interests by including the following statement: ""This does not alter our adherence to  PLOS ONE policies on sharing data and materials.” If there are restrictions on sharing of data and/or materials, please state these. Please note that we cannot proceed with consideration of your article until this information has been declared.

Reviewers' comments:

Reviewer's Responses to Questions

**Comments to the Author**

1. Is the manuscript technically sound, and do the data support the conclusions?

Reviewer #1: Yes

Reviewer #2: Yes

2. Has the statistical analysis been performed appropriately and rigorously? 

Reviewer #1: N/A

Reviewer #2: N/A

3. Have the authors made all data underlying the findings in their manuscript fully available?

Reviewer #1: Yes

Reviewer #2: Yes

4. Is the manuscript presented in an intelligible fashion and written in standard English?

Reviewer #1: Yes

Reviewer #2: Yes

5. Review Comments to the Author

Reviewer #1: Comments:

The authors developed an Escherichia coli strain named MG1655-ISC to produce seamless DNA vectors. In a further step, they gave two examples for its applications. It could be used as a useful method to prepare minimized vectors, though the innovation of this method is not high. Both the preparation of minicircles by using λ integrase and the development of IntC3 has been reported by the authors in other papers (Shree, et al., 2016, Nucleic Acids Research; Namrata Chaudhari, et al., Stem Cell Res Ther.). It is not hard to combine these two ideas together for method optimization. Anyway, the study did present the results of primary scientific research, and the results sound credible. I think it fits to basic criteria of Plos One. If my understanding about the criteria is right, the manuscript could be acceptable after the concerned issues have reasonable explanations.

Some concerns:

1. Line 53-54: ‘However, employing these systems reproducibly and at larger scales remains technically challenging.’ Could the authors give further explanations on the challenges? I could not go there basing on the information provided by the authors.

2. Line 61: Could the author give enough features on the ‘flexibility in att sites’? It is the main ‘novelty’ of this method and would be quite useful for other people to design their DNA sequences basing on this method.

3. Line 70: I do not know the relationship between the tightly regulated expression of ara operon and minimal prior genetic changes. Could the authors give some proofs or references?

4. Line 73: The genome of MG1655 has some gene mutations except λ- and F-. ‘K-12 F– λ– ilvG– rfb-50 rph-1’ This genotype feature is derived from website of openwetware.

5. In the result part, the authors also used the protein scIHF2. I suggested the authors also described its background and purpose in the introduction part.

6. The sequences termed ISC was not long enough in Fig. S1. I also suggested the authors marked the names of each gene or element with underlines.

7. I suggested the authors could give a detailed protocol in the method section besides the development of MG1655-ISC.

8. After the authors got the minicircles in the enzyme reaction buffer, dose it necessary to be purified again or direct to use with these complex enzymes and reagents? If purification is necessary, does the authors calculate the yield before purification or after purification? In addition, I do not know how to calculate minicircles yield in the process. Could the authors give detailed calculation process? (Line 211) In further, if purification is necessary, I also suggested the author gave a gel picture in Fig. 3 and Fig. 4 to show if it is possible the get ‘clean’ DNA in the two examples.

9. Line 203: From lane 4 and lane 5 in Fig. 4, if the ‘majority’ of the DNA recombined, there should be only two bands. However, there are four bands with similar luminance. Majority seems to be improper. If my understandingis right, that would indicate the recombination mediated by IntC3 in cells is not thorough as expected. Could the author give some reasons or discuss it in the discussion section?

Reviewer #2: The researchers in this study described the engineering and application of a novel E. coli strain (MG1655-ISC) carrying a tightly regulated gene cassette for the expression of the enhanced mutant lambda integrase variant IntC3 inserted into the ara operon. The authors have established a simple and efficient cell culture protocol demonstrating IntC3 very efficiently recombined plasmids inside E. coli to easily produce different scales of seamless vectors ranging in size between 0.5 kb and >10 kb. The employment of restriction enzymes together with the highly active phage exonuclease T5 ultimately resulted in high yields of pure negatively supercoiled seamless vectors. This study has potential to expand the scope of future downstream applications for seamless vectors including site-specific genome editing of higher eukaryotic cells.

The manuscript is nicely written and the work is well conducted with appropriate controls. I recommend the paper could be published in the “PloS One” after the authors address the below minor points.

Minor points:

1. Line 116: The authors mentioned “degrees” instead of its symbol as used elsewhere in the manuscript.

2. Line 162, 179, 181, 204: “digestion” instead of “digest”.

3. Line 191: Add a space between “13.4” and “kb”.

4. Line 198: “was analyzed” instead of “analyzed”.

5. Units at some places have a space between number and units and some places do not have it. Please maintain uniformity throughout the manuscript.

6. At some places, the authors have written “lambda integrase” and at some places with the symbol lambda. Please maintain uniformity throughout the manuscript. Check for similar errors and edit it.

6. PLOS authors have the option to publish the peer review history of their article (what does this mean?). If published, this will include your full peer review and any attached files.

Reviewer #1: **Yes: **Yongzhen Xia

Reviewer #2: No

---

## [Author Response · Author response to Decision Letter 0]

3 Sep 2022

Response to Reviewers

We would like to thank the reviewers for his/her thoughtful comments and efforts towards improving the manuscripts. 

In the following sections, we address each point raised by the reviewers. All page numbers mentioned refer to the amended manuscript.

Reviewer #1: Comments:

1. Line 53-54: ‘However, employing these systems reproducibly and at larger scales remains technically challenging.’ Could the authors give further explanations on the challenges? I could not go there basing on the information provided by the authors.

We have included the explanation in the amended manuscript in line 54-56. 

2. Line 61: Could the author give enough features on the ‘flexibility in att sites’? It is the main ‘novelty’ of this method and would be quite useful for other people to design their DNA sequences basing on this method.

“att” sites are flexible because not only attP but also attL can also recombine with attH for site specific recombination in human genome (Chandra et al., 2016; Makhija et al., 2018). Thus, the parental plasmids can be designed with different combinations of att sites for seamless vector production 

3. Line 70: I do not know the relationship between the tightly regulated expression of ara operon and minimal prior genetic changes. Could the authors give some proofs or references?

Since MG1655 is cured of temperate lambda bacteriophage, hence there will be no leaky expression of integrase from the integrated lambda prophage (mentioned in line 73-75). In addition, as stated, we reasoned those minimal genetic changes will contribute to the functionality of the ara operon. We amended this in line 71 to “hypothesized”.

4. Line 73: The genome of MG1655 has some gene mutations except λ- and F-. ‘K-12 F– λ– ilvG– rfb-50 rph-1’ This genotype feature is derived from website of openwetware.

The mutations listed in the genotype are present in most K-12 strains and were probably acquired early in the history of the laboratory strain. A frameshift at the end of rph results in decreased pyrE expression and a mild pyrimidine starvation, such that the strain grows 10 to 15% more slowly in pyrimidine-free medium than in medium containing uracil (Jensen, 1993). The ilvG- mutation is a frameshift that knocks out acetohydroxy acid synthase II (Lawther et al., 1982). The rfb-50 mutation is an IS5 insertion that results in the absence of O-antigen synthesis (Liu and Reeves, 1994).

5. In the result part, the authors also used the protein scIHF2. I suggested the authors also described its background and purpose in the introduction part.

We have followed this suggestion and incorporated the background of protein scIHF2 in the amended manuscript in line 78-80. 

6. The sequences termed ISC was not long enough in Fig. S1. I also suggested the authors marked the names of each gene or element with underlines.

We have followed this suggestion and incorporated this change in Figure S1.

7. I suggested the authors could give a detailed protocol in the method section besides the development of MG1655-ISC.

The detailed protocol for seamless vector preparation has been already mentioned in line 154-169. 

8. After the authors got the minicircles in the enzyme reaction buffer, dose it necessary to be purified again or direct to use with these complex enzymes and reagents? If purification is necessary, does the authors calculate the yield before purification or after purification? In addition, I do not know how to calculate minicircles yield in the process. Could the authors give detailed calculation process? (Line 211) In further, if purification is necessary, I also suggested the author gave a gel picture in Fig. 3 and Fig. 4 to show if it is possible the get ‘clean’ DNA in the two examples.

After restriction digestion and exonuclease treatment, minicircles are purified by phenol/chloroform extraction and ethanol precipitation. Aliquots of purified DNA are quantified by standard OD readings and total yields calculated accordingly. We have added a note in line 218. The purified seamless vector was shown in lane no. 5 & 6 of Figure 3 (B) and lane no. 6 of Figure 4 (B). 

9. Line 203: From lane 4 and lane 5 in Fig. 4, if the ‘majority’ of the DNA recombined, there should be only two bands. However, there are four bands with similar luminance. Majority seems to be improper. If my understanding is right, that would indicate the recombination mediated by IntC3 in cells is not thorough as expected. Could the author give some reasons or discuss it in the discussion section?

The combined intensity of the two expected recombination products (band 10.3kb for linear seamless vector and 3.1kb for linear bacterial backbone) is much higher than the combined intensities of unrecombined plasmid (bands 7.8kb and 5.6kb from unrecombined plasmid). Hence, we stated that the majority of the substrate plasmid had undergone recombination. 

The figure legend for Figure 4 has been amended in line 310-319.

Reviewer #2: Comments:

1. Line 116: The authors mentioned “degrees” instead of its symbol as used elsewhere in the manuscript.

We have followed this suggestion and incorporated the change in the amended manuscript in line 115. 

2. Line 162, 179, 181, 204: “digestion” instead of “digest”.

We have followed this suggestion and incorporated the change in the amended manuscript in Line 162, 179, 181, 204.

3. Line 191: Add a space between “13.4” and “kb”.

We have followed this suggestion and incorporated the change in the amended manuscript in line 191. 

4. Line 198: “was analyzed” instead of “analyzed”.

We have followed this suggestion and incorporated the change in the amended manuscript in line 198. 

5. Units at some places have a space between number and units and some places do not have it. Please maintain uniformity throughout the manuscript.

We have followed this suggestion and incorporated the change in the amended manuscript. 

6. At some places, the authors have written “lambda integrase” and at some places with the symbol lambda. Please maintain uniformity throughout the manuscript. Check for similar errors and edit it.

We have followed this suggestion and incorporated the change in the amended manuscript. 

References: 

K F Jensen. The Escherichia coli K-12 "Wild Types" W3110 and MG1655 Have an rph Frameshift Mutation That Leads to Pyrimidine Starvation Due to low pyrE Expression Levels. J Bacteriol. 1993; 175(11): 3401–3407. doi: 10.1128/jb.175.11.3401-3407. PubMed PMID: 8501045.

Liu D and Reeves PR. Escherichia coli K12 regains its O antigen. Microbiol. 1994; 140: 49-57. doi.org/10.1099/13500872-140-1-49. PubMed PMID: 7512872.

Makhija H, Roy S, Hoon S, Ghadessy FJ, Wong D, Jaiswal R, et al. A novel lambda integrase-mediated seamless vector transgenesis platform for therapeutic protein expression. Nucleic Acids Res. 2018;46(16):e99. Epub 2018/06/13. doi: 10.1093/nar/gky500. PubMed PMID: 29893931; PubMed Central PMCID: PMCPMC6144826

R P Lawther, D H Calhoun, J Gray, C W Adams, C A Hauser, G W Hatfield. DNA Sequence Fine-Structure Analysis of ilvG (IlvG+) Mutations of Escherichia coli K-12. J Bacteriol. 1982 Jan; 149(1): 294–298. doi: 10.1128/jb.149.1.294-298.1982. PubMed PMID: 7033211.

Vijaya Chandra SH, Makhija H, Peter S, Myint Wai CM, Li J, Zhu J, et al. Conservative site-specific and single-copy transgenesis in human LINE-1 elements. Nucleic Acids Res. 2016;44(6):e55. Epub 2015/12/18. doi: 10.1093/nar/gkv1345. PubMed PMID: 26673710; PubMed Central PMCID: PMCPMC4824084.

---

## [Decision Letter · Decision Letter 1]

8 Sep 2022

Versatile seamless DNA vector production in E. coli using enhanced phage lambda integrase

PONE-D-22-16259R1

Dear Dr. Dröge,

We’re pleased to inform you that your manuscript has been judged scientifically suitable for publication and will be formally accepted for publication once it meets all outstanding technical requirements.

Kind regards,

Chen Ling, Ph.D.

Academic Editor

PLOS ONE

Additional Editor Comments (optional):

Reviewers' comments:

Reviewer's Responses to Questions

**Comments to the Author**

1. If the authors have adequately addressed your comments raised in a previous round of review and you feel that this manuscript is now acceptable for publication, you may indicate that here to bypass the “Comments to the Author” section, enter your conflict of interest statement in the “Confidential to Editor” section, and submit your "Accept" recommendation.

Reviewer #1: All comments have been addressed

Reviewer #2: All comments have been addressed

2. Is the manuscript technically sound, and do the data support the conclusions?

Reviewer #1: Yes

Reviewer #2: Yes

3. Has the statistical analysis been performed appropriately and rigorously? 

Reviewer #1: N/A

Reviewer #2: N/A

4. Have the authors made all data underlying the findings in their manuscript fully available?

Reviewer #1: Yes

Reviewer #2: Yes

5. Is the manuscript presented in an intelligible fashion and written in standard English?

Reviewer #1: Yes

Reviewer #2: Yes

6. Review Comments to the Author

Reviewer #1: Reviewer comments

Most of the questions have been solved, but there are still some that need to be further revised.

1. Line 61: Could the author give enough features on the ‘flexibility in att sites’? It is the main ‘novelty’ of this method and would be quite useful for other people to design their DNA sequences basing on this method.

“att” sites are flexible because not only attP but also attL can also recombine with attH for site specific recombination in human genome (Chandra et al., 2016; Makhija et al., 2018). Thus, the parental plasmids can be designed with different combinations of att sites for seamless vector production

More suggestions: I suggested that the author put this information in the introduction section.

2. I suggested the authors could give a detailed protocol in the method section besides the development of MG1655-ISC.

The detailed protocol for seamless vector preparation has been already mentioned in line 154-169.

More suggestions: I suggested the authors to describe how to prepare the seamless vector in more detailed manner. First, the volume to culture the cells could be given, since I would like to know the scale. Because the author has declared ‘large scales’ is a problem (line 54). I would like to know if the author could solve this problem here. Second, the steps about phenol/chloroform extraction and ethanol precipitation (line 218) or declared ‘standard procedures’ here (line 166) should be given too. At least a reference could be given to let audience follow. Third, how could the concentration of DNA be determined by using a standard OD method?

A more confusing thing is that how to treat the DNA with restriction digestion and T5 exonuclease in follow? Should we purify them before T5 digestion, or should we just direct add T5 enzyme inside? Do we need to purify the DNA several times with the same procedure?

In addition, I do not think it is proper to describe ‘method’ in the result section. Principle of the method could be given in the result section, but not detailed parameters.

Reviewer #2: The authors have satisfactorily addressed all the comments raised by the reviewers and can be recommended for publication.

7. PLOS authors have the option to publish the peer review history of their article (what does this mean?). If published, this will include your full peer review and any attached files.

Reviewer #1: No

Reviewer #2: No

---

## [Editor Report · Acceptance letter]

16 Sep 2022

PONE-D-22-16259R1 

Versatile seamless DNA vector production in E. coli using enhanced phage lambda integrase 

Dear Dr. Dröge:

I'm pleased to inform you that your manuscript has been deemed suitable for publication in PLOS ONE. Congratulations! Your manuscript is now with our production department. 

Kind regards, 

on behalf of

Dr. Chen Ling 

Academic Editor

PLOS ONE